# Genome-wide association study identifies susceptibility loci for B-cell childhood acute lymphoblastic leukemia

Jayaram Vijayakrishnan[1], James Studd [1], Peter Broderick [1], Ben Kinnersley [1], Amy Holroyd[1], Philip J. Law [1], Rajiv Kumar[2], James M. Allan[3], Christine J. Harrison[4], Anthony V. Moorman[4], Ajay Vora[5], Eve Roman[6], Sivaramakrishna Rachakonda[2], Sally E. Kinsey[7], Eamonn Sheridan[8], Pamela D. Thompson[9], Julie A. Irving[3], Rolf Koehler[10], Per Hoffmann[11,12], Markus M. Nöthen[11], Stefanie Heilmann-Heimbach[11], Karl-Heinz Jöckel[13], Douglas F. Easton [14,15], Paul D.P. Pharoah [14,15], Alison M. Dunning[16], Julian Peto[17], Frederico Canzian [18], Anthony Swerdlow[1,19], Rosalind A. Eeles [1,20], ZSofia Kote-Jarai[1], Kenneth Muir [21,22], Nora Pashayan [15,23], The PRACTICAL consortium, Mel Greaves[24], Martin Zimmerman[25], Claus R. Bartram[10], Martin Schrappe[26], Martin Stanulla[25], Kari Hemminki[2,27] & Richard S. Houlston[1]

Genome-wide association studies (GWAS) have advanced our understanding of susceptibility to B-cell precursor acute lymphoblastic leukemia (BCP-ALL); however, much of the heritable risk remains unidentified. Here, we perform a GWAS and conduct a meta-analysis with two existing GWAS, totaling 2442 cases and 14,609 controls. We identify risk loci for BCP-ALL at 8q24.21 (rs28665337, $P = 3.86 \times 10^{-9}$, odds ratio (OR) = 1.34) and for *ETV6-RUNX1* fusion-positive BCP-ALL at 2q22.3 (rs17481869, $P = 3.20 \times 10^{-8}$, OR = 2.14). Our findings provide further insights into genetic susceptibility to ALL and its biology.

[1] Division of Genetics and Epidemiology, The Institute of Cancer Research, Sutton, Surrey SM2 5NG, UK. [2] Division of Molecular Genetic Epidemiology, German Cancer Research Centre, 69120 Heidelberg, Germany. [3] Northern Institute for Cancer Research, Newcastle University, Newcastle upon Tyne NE2 4HH, UK. [4] Wolfson Childhood Cancer Research Centre, Northern Institute for Cancer Research, Newcastle University, Newcastle upon Tyne NE1 7RU, UK. [5] Department of Haematology, Great Ormond Street Hospital, London WC1N 3JH, UK. [6] Department of Health Sciences, University of York, York YO10 5DD, UK. [7] Department of Paediatric and Adolescent Haematology and Oncology, Leeds General Infirmary, Leeds LS1 3EX, UK. [8] Medical Genetics Research Group, Leeds Institute of Molecular Medicine, University of Leeds, Leeds LS9 7TF, UK. [9] Paediatric and Familial Cancer Research Group, Institute of Cancer Sciences, St. Mary's Hospital, Manchester M13 9WL, UK. [10] Department of Human Genetics, Institute of Human Genetics, University of Heidelberg, 69120 Heidelberg, Germany. [11] Department of Genomics, Institute of Human Genetics, Life & Brain Centre, University of Bonn, D-53012 Bonn, Germany. [12] Department of Biomedicine, Human Genomics Research Group, University Hospital and University of Basel, 4031 Basel, Switzerland. [13] Institute for Medical Informatics, Biometry and Epidemiology, University Hospital Essen, University of Duisburg–Essen, 45147 Essen, Germany. [14] Department of Oncology, Centre for Cancer Genetic Epidemiology, University of Cambridge, Cambridge CB1 8RN, UK. [15] Department of Public Health and Primary Care, Centre for Cancer Genetic Epidemiology, University of Cambridge, Cambridge CB1 8RN, UK. [16] Department of Oncology, Centre for Cancer Genetic Epidemiology, University of Cambridge, Strangeways Laboratory, Cambridge CB1 8RN, UK. [17] Department of Non-Communicable Disease Epidemiology, London School of Hygiene and Tropical Medicine, London WC1E 7HT, UK. [18] Genomic Epidemiology Group, German Cancer Research Center (DKFZ), 69120 Heidelberg, Germany. [19] Division of Breast Cancer Research, The Institute of Cancer Research, London SW7 3RP, UK. [20] Royal Marsden NHS Foundation Trust, London SW3 6JJ, UK. [21] Institute of Population Health, University of Manchester, Manchester M13 9PL, UK. [22] Warwick Medical School, University of Warwick, Coventry CV4 7AL, UK. [23] Department of Applied Health Research, University College London, London WC1E 7HB, UK. [24] Centre for Evolution and Cancer, Institute of Cancer Research, Sutton, Surrey SM2 5NG, UK. [25] Department of Paediatric Haematology and Oncology, Hannover Medical School, 30625 Hannover, Germany. [26] General Paediatrics, University Hospital Schleswig-Holstein, 24105 Kiel, Germany. [27] Center for Primary Health Care Research, Lund University, 221 00 Lund, Sweden. Correspondence and requests for materials should be addressed to R.S.H. (email: richard.houlston@icr.ac.uk).
#Full list of consortium members appears at the end of the paper.

A cute lymphoblastic leukemia (ALL) is the most common pediatric cancer in western countries, of which B-cell precursor acute lymphoblastic leukemia (BCP-ALL) accounts for approximately 80% of cases[1]. The etiology of ALL is poorly understood and no specific environmental risk factor has so far been identified aside from indirect evidence for an infective origin[2,3]. Independent of concordance disease in monozygotic twins, which has an in utero origin evidence, albeit indirect, for inherited predisposition to ALL is provided by the elevated risk seen in siblings of ALL cases[4]. Previous genome-wide association studies (GWAS)[5–9] have suggested susceptibility to ALL is polygenic, identifying single-nucleotide polymorphisms (SNPs) in eight loci influencing ALL risk at 7p12.2 (*IKZF1*), 9p21.3 (*CDKN2A*), 10p12.2 (*PIP4K2A*), 10q26.13 (*LHPP*), 12q23.1 (*ELK3*), 10p14 (*GATA3*), 10q21.2 (*ARID5B*), and 14q11.2 (*CEBPE*). ALL is biologically heterogeneous and subtype associations have been identified for 10q21.2 (*ARID5B*) associated with high-hyperdiploid BCP-ALL (i.e., >50 chromosomes) and 10p14 (*GATA3*) associated with Ph-like BCP-ALL[6,10].

Statistical modeling of GWAS data indicates that much of the heritable risk of ALL ascribable to common genetic variation remains to be discovered[5–9]. To gain a more comprehensive insight into predisposition to ALL we performed a meta-analysis of two previously published GWAS and a new GWAS together totaling 2442 cases and 14,609 controls. We report two previously unidentified risk loci, providing further insights into the genetic and biological basis of this disease.

## Results

**Association analysis**. We analyzed data from three studies of European ancestry: a new GWAS from the United Kingdom–UK GWAS II, and two previously reported GWAS–UK GWAS I and a German GWAS (Supplementary Figs. 1, 2 and Supplementary Table 1). After imposing pre-determined (see "Methods") quality metrics to each of the three GWAS, the studies provided genotype data on 2442 cases and 14,609 controls. To increase genomic resolution, we imputed >10 million SNPs using whole-genome reference genotype data from 1000 Genomes Project ($n = 1092$)[11] and UK10K ($n = 3781$)[12]. Quantile-quantile plots of SNPs (minor allele frequency (MAF) > 0.01) post-imputation showed no evidence of substantive over-dispersion introduced by imputation (genomic inflation[13] $\lambda$ for UK GWAS I, UK GWAS II, and German GWAS were 1.02, 1.05, and 1.01, respectively; Supplementary Fig. 3)[6,7].

Pooling data from the three GWAS, we derived joint odds ratios (ORs), 95% confidence intervals (CIs), and associated per allele P-values under a fixed-effects model for each SNP with MAF > 0.01. Given the biological heterogeneity of BCP-ALL, overall and subtype-specific ORs were derived for BCP-ALL, high-hyperdiploid ALL (i.e., >50 chromosomes), and *ETV6-RUNX1* fusion-positive BCP-ALL. This combined meta-analysis further substantiated previously published risk SNPs (Fig. 1, Supplementary Table 2). In addition to previously reported loci we identified three risk loci for BCP-ALL at 8q24.21 (rs28665337, hg19 chr8:g.130194104) and 5q21.3 (rs7449087, hg19 chr5: g.107928071), and for *ETV6-RUNX1*-positive ALL at 2q22.3 (rs17481869, hg19 chr2:g.146124454) (Fig. 2, Tables 1 and 2, Supplementary Table 3). rs17481869 was genotyped in UK GWAS II and German GWAS, while rs28665337 was imputed (info score > 0.97) in all three data sets, imputation fidelity was confirmed through Sanger sequencing in a subset of samples ($r^2 = 0.98$, Supplementary Table 4). The fidelity of imputation of SNP rs7449087 was poor ($r^2 = 0.81$) with no correlated directly typed SNP with P-value < $1 \times 10^{-6}$, hence we did not consider this represented a bona fide association (Supplementary Table 4). Conditional analysis did not provide evidence for multiple independent signals at either 8q24.21 or 2q22.3.

The 8q24.21 variant rs28665337 maps 35 kb 3′ of the long intergenic non-coding RNA 977 (*LINC00977*, Fig. 2). The 8q24.21 region harbors variants associated with multiple cancers, including colorectal, prostate, bladder cancer also B-cell malignancies such as diffuse large B-cell lymphoma, Hodgkin lymphoma, and chronic lymphocytic leukemia (Supplementary Table 5). The linkage disequilibrium (LD) blocks delineating these cancer risk loci are distinct from the 8q24.21 BCP-ALL association signal suggesting this risk locus is unique to BCP-ALL (pairwise LD metrics $r^2 < 0.2$; Supplementary Table 5). rs17481869 maps to an intergenic region at 2q22.3 with no candidate gene nearby (Fig. 2).

**Relationship between SNP genotype and patient outcome**. We examined the relationship between SNP genotype and patient outcome using data from UK GWAS II and German GWAS. Neither rs28665337 or rs17481869 showed a consistent association with either event-free survival (EFS) or risk of relapse, even when stratified by *ETV6-RUNX1* status (Supplementary Table 6).

**Functional annotation of risk loci**. To gain insight into the biological basis underlying the association signals at these as well as previously identified risk loci, we examined the epigenetic landscape of BCP-ALL risk loci genome wide. For each risk locus we evaluated profiles of three histone marks of active chromatin

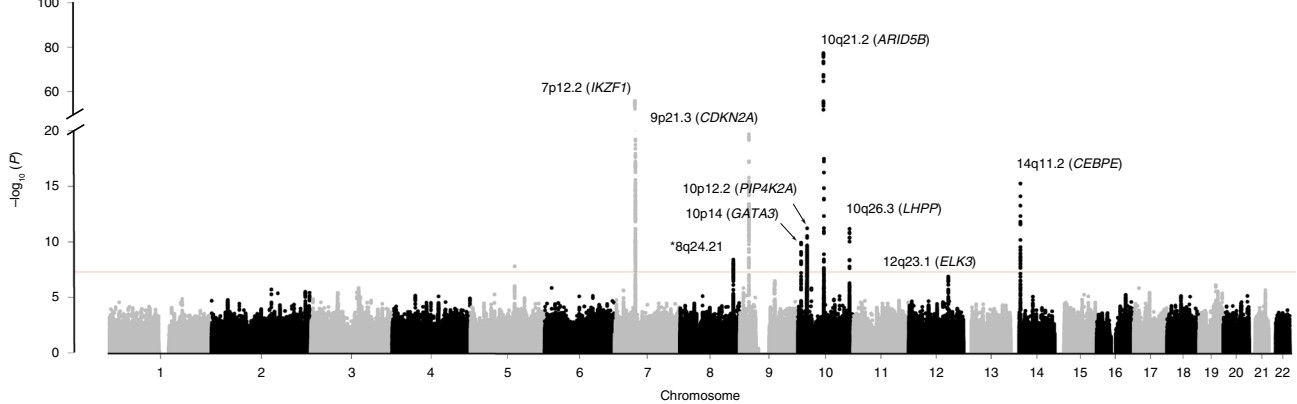

**Fig. 1** Manhattan plot of association. y-axis shows genome-wide P-values (two-sided, calculated using SNPTEST v2.5.2 assuming an additive model) of >6 million successfully imputed autosomal SNPs in 2442 cases and 14,609 controls. The x-axis shows the chromosome number. The red horizontal line represents the genome-wide significance threshold of $P = 5.0 \times 10^{-8}$

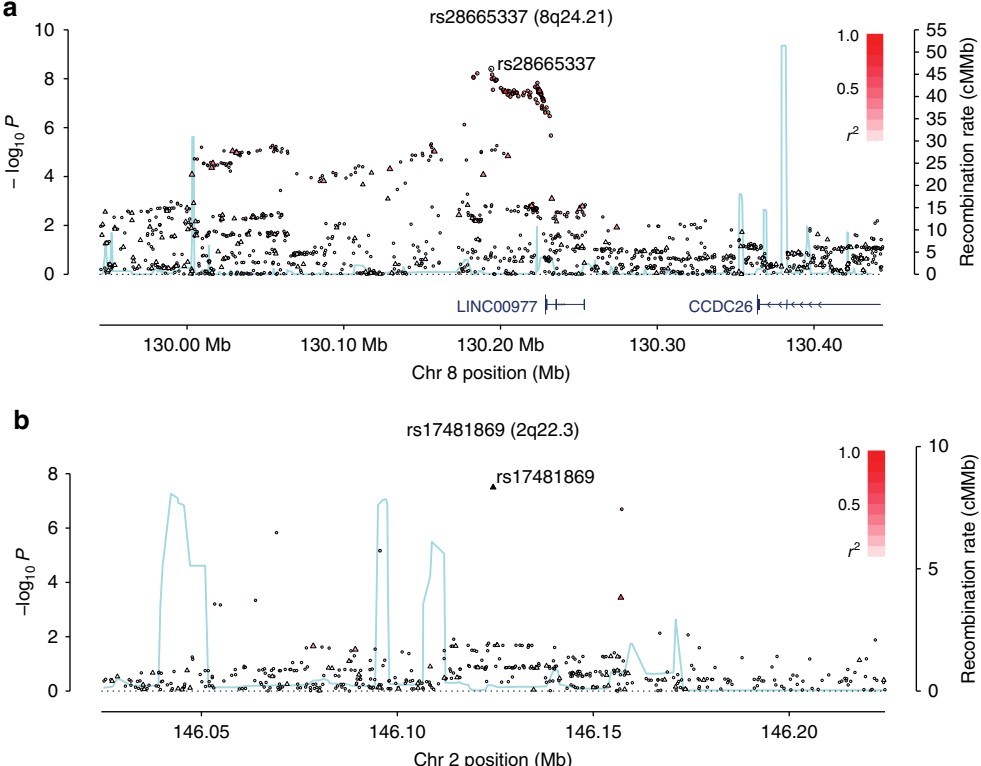

**Fig. 2** Regional plots of association results and recombination rates for the identified risk loci. **a** 8q24.21 (rs28665337), **b** 2q22.3 (rs17481869). Plots (generated using visPIG[14]) show association $-\log_{10}P$-values (left $y$-axis) of genotyped (triangles) and imputed (circles) SNPs in the GWAS samples (2442 cases and 14,609 controls) and recombination rates (right $y$-axis). $-\log_{10}P$-values were calculated assuming an additive model in SNPTEST v2.5.2 and are shown according to their chromosomal positions ($x$-axis). Lead SNPs are denoted by large circles or triangles labeled by rsID. Color intensity of each symbol reflects LD, white ($r^2 = 0$), dark red ($r^2 = 1.0$). Light blue line shows recombination rates from UK10K Genomes Project. Genome coordinates are from NCBI human genome GRCh37

(H3K27ac, H3K4me1, and H3k4me3) using ChIP-seq data of 14 cell types from ENCODE, including lymphoblastoid cell line (GM12878), and multiple ALL and acute myeloid leukemia (AML) samples from the Blue-Print Epigenome database (Supplementary Fig. 4, Supplementary Table 7)[15,16]. Since the strongest associated GWAS SNP may not represent the causal variant, we examined signals across an interval spanning all variants in LD with the most strongly associated SNP at each risk loci ($r^2 >$ 0.8 and $D' > 0.8$ based on the 1000 Genomes EUR reference panel). The analysis across all risk loci combined revealed that risk SNPs are enriched for markers of open chromatin and that enrichment is highest in ALL cells (Supplementary Fig. 4, Supplementary Table 7). Analysis using HaploReg[17] revealed a significant enrichment of SNPs within enhancers in primary hematopoietic stem cells (binomial test for enrichment, $P =$ 0.0034; Supplementary Data 1). Collectively these data support a model of disease etiology where risk loci influence BCP-ALL risk through *cis* regulatory effects on transcription.

We used summary-level Mendelian randomization (SMR) analysis to test for concordance between GWAS and *cis*-eQTL-associated SNPs with all correlated SNPs ($r^2 > 0.8$) within 1 Mb of the lead SNP at each locus (Supplementary Tables 8 and 9) deriving $b_{XY}$ statistics, which estimate the effect of gene expression on childhood ALL risk. This analysis showed variation in the expression of *CDKN2B*, *FAM53B*, *FIGNL1*, and *PIP5K2A* were associated with risk loci (Supplementary Fig. 5, Supplementary Tables 8 and 9). Eight gene probes exceeded the $P_{SMR}$ threshold of $1.3 \times 10^{-4}$, of which two genes passed the HEIDI test for heterogeneity ($P_{HEIDI} > 0.05$). In whole blood-derived tissue, the 10q26.13 locus was associated with *FAM53B* expression and

the 10p12.2 locus was associated with *PIP4K2A* (alias *PIP5K2A*) expression ($P_{SMR} = 2.09 \times 10^{-4}$, $b_{xy} = -0.99$, and $P_{SMR} = 7.48 \times 10^{-8}$, $b_{xy} = 0.32$, respectively; Supplementary Fig. 5, Supplementary Table 9). Following from SMR analysis we also investigated whether the most strongly associated SNP at each risk locus, individually, was associated with the expression of genes within a 2 MB window to ensure capture of long range interactions. This provided evidence for a relationship between the 8q24.21 risk allele (rs28665337) and increased expression of *MYC* ($t$-test, $P = 7.20 \times 10^{-4}$; Supplementary Fig. 6, Supplementary Table 10), and the 2q22.3 risk allele (rs17481869) with decreased *GTDC1* expression ($t$-test, $P = 0.037$; Supplementary Fig. 6, Supplementary Table 10). Since chromatin looping interactions are fundamental for regulation of gene expression, we interrogated physical interactions at respective genomic regions defined by rs28665337 and rs17481869 in GM12878 lymphoblastoid and H1 human embryonic stem (ES) cells using Hi-C data. Acknowledging limitations that these cell types may not fully reflect ALL biology, the regions containing rs28665337 and rs17481869 show significant chromatin looping interactions with the promotor regions of *MYC* in ES cells and *GTDC1* in GM12878, respectively (Fit-Hi-C test[18], Supplementary Figs. 7, 8).

**HLA alleles and risk**. A relationship between variation within the major histocompatibility complex (MHC) region and risk of ALL has long been speculated[19–26]. However, most studies have failed to address the complex LD patterns within the MHC or issues relating to population stratification. In view of the inconsistencies and limitations of published studies we conducted a more rigorous

**Table 1 rs28665337 (8q24.21) genotypes and risk associated with BCP-ALL, high-hyperdiploid, and *ETV6-RUNX1*-positive childhood BCP-ALL subtypes**

| | RAF | | Number | | | | |
| --- | --- | --- | --- | --- | --- | --- | --- |
| **All BCP-ALL** | **Cases** | **Controls** | **Cases** | **Controls** | **OR** | **CI** | **P-value** |
| UK GWAS I | 0.15 | 0.12 | 824 | 5200 | 1.32 | (1.12–1.55) | $7.91 \times 10^{-4}$ |
| German GWAS | 0.16 | 0.12 | 834 | 2024 | 1.28 | (1.07–1.53) | $7.64 \times 10^{-3}$ |
| UK GWAS II | 0.15 | 0.12 | 784 | 7385 | 1.39 | (1.21–1.47) | $4.16 \times 10^{-5}$ |
| Combined | | | 2442 | 14,609 | 1.34 | (1.21–1.47) | $3.86 \times 10^{-9}$ |
| | | | | | | $P_{het} = 0.77$ | $I^2 = 0\%$ |
| **High-hyperdiploid** | | | | | | | |
| UK GWAS I | 0.15 | 0.12 | 289 | 5200 | 1.45 | (1.11–1.88) | $6.30 \times 10^{-3}$ |
| German GWAS | 0.17 | 0.12 | 176 | 2024 | 1.49 | (1.06–2.09) | $2.29 \times 10^{-2}$ |
| UK GWAS II | 0.15 | 0.12 | 251 | 7385 | 1.38 | (1.05–1.81) | $2.19 \times 10^{-2}$ |
| Combined | | | 716 | 14,609 | 1.49 | (1.21–1.87) | $2.55 \times 10^{-5}$ |
| | | | | | | $P_{het} = 0.94$ | $I^2 = 0\%$ |
| ***ETV6-RUNX1*-positive** | | | | | | | |
| UK GWAS I | 0.16 | 0.12 | 126 | 5200 | 1.51 | (1.01–2.26) | $4.27 \times 10^{-2}$ |
| German GWAS | 0.09 | 0.12 | 63 | 2024 | 0.78 | (0.44–1.38) | $3.93 \times 10^{-1}$ |
| UK GWAS II | 0.14 | 0.12 | 220 | 7385 | 1.23 | (0.94–1.62) | $1.38 \times 10^{-1}$ |
| Combined | | | 409 | 14,609 | 1.23 | (1.00–1.51) | $5.20 \times 10^{-4}$ |
| | | | | | | $P_{het} = 0.18$ | $I^2 = 42\%$ |

Note: *P*-values for each individual study were generated using SNPTEST v2.5.2 software. Combined *P*-values and estimates were obtained using a fixed-effects model using beta values and standard errors. *RAF* risk allele frequency, *OR* odds ratio, $P_{het}$ *P* heterogeneity, $I^2$ index to quantify dispersion of odds ratio, *CI* confidence interval

analysis. Specifically, we investigated a possible relationship between BCP-ALL risk and HLA alleles by imputing the 6p21 region using the Type I Diabetes Genetics Consortium (T1DGC) as reference[27–29]. The strongest association from a combined analysis of all three GWAS was provided by SNP rs9469021, which maps 167 Kb centromeric to HLA-B (combined $P = 3.5 \times 10^{-3}$; frequentist test of association using SNPTEST); this association was, however, not significant after correcting for multiple testing.

**Impact on heritable risk**. Using genome-wide complex trait analysis (GCTA)[30–32] the heritability of BCP-ALL accounted for by common variants was estimated to be 0.16 (±standard error (S.E.) 0.03, REML analysis $P_{meta} = 4.25 \times 10^{-8}$) with little evidence for subtype difference (0.18 ± S.E. 0.05 and 0.20 ± S.E. 0.08 for hyperdiploid and *ETV6-RUNX1*-positive BCP-ALL, respectively). The 11 known susceptibility variants account for 34% of the familial risk (Supplementary Table 11). The impact of BCP-ALL SNPs are among the strongest GWAS associations of any malignancy, raising the possibility of clinical utility for risk prediction. To examine this, we generated polygenic risk scores (PRS) based on the composite effect of all risk SNPs assuming a log-normal relative risk distribution. Using this approach for all risk SNPs, individuals in the top 1% of genetic risk had a 7.5-fold relative risk of BCP-ALL (Supplementary Fig. 9). The individual risk discrimination provided by the variants is shown in the receiver–operator characteristic (ROC) curves with the area under the curve (AUC) being 0.73 (Supplementary Fig. 10).

## Discussion

The evidence for the two risk loci we report has been based on a meta-analysis of three independent GWAS data sets. While the combined association *P*-values for each risk locus is genome-wide significant with each series providing support for association we acknowledge that we did not provide additional replication. For rare cancers such as childhood ALL, ascertaining case series which are appropriately ethnically matched and are sufficiently powered to provide independent replication is inherently problematic. Moreover as exemplified by the 10q21 and 10p14 risk

loci, associations can be highly subtype-specific which adds to the difficulty in obtaining appropriate replication series. Accepting such caveats our analysis provides evidence for the existence of two additional risk loci for childhood BCP-ALL at 2q22.3 and 8q24.21.

We did not observe an association between risk SNPs at either 2q22.3 and 8q24.21 with patient survival. This is consistent with the impact of risk variants operating at an early stage of ALL evolution rather than disease progression per se. We acknowledge this analysis only has power to demonstrate a 10% difference in patient outcome. To robustly determine the relationship between genotype and outcome requires larger patient cohorts.

Given the existence of different subtypes of BCP-ALL, presumably reflecting the different etiology and evolutionary trajectories, it is perhaps not surprising that some SNPs display subtype-specific effects. Notable in this respect are the 10q21.2 and 10p14 variants that specifically influence high-hyperdiploid BCP-ALL[33] and Ph-like ALL[10], respectively. As with 7p12.2, 9p21.3, 10p12.2, 14q11.2, and the currently identified 8q24.21 locus has generic effects on the risk of BCP-ALL. In contrast the 2q22.3 association was highly specific for *ETV6-RUNX1*-positive BCP-ALL.

Deregulation of *MYC* has been reported in ALL, in some instances as a consequence of chromosomal rearrangement[34]. Studies in other cancers have shown that disease-specific risk loci at 8q24.21 lie within tissue-specific enhancers interacting with *MYC* or *PVT1* promotors. Furthermore, recent Hi-C analysis of this region has demonstrated a complicated 3D structure implicating various lncRNAs in mediating risk[35]. Hence, it is plausible that the susceptibility to ALL has a similar mechanistic basis, brought about through involvement of the lincRNA 00977.

Risk conferred by rs17481869 (2q22.3) was specific to *ETV6-RUNX1*-positive BCP-ALL. The SNP association is intergenic with no obvious candidate gene in the vicinity, presently hindering the suggestion of testable hypotheses regarding its functional basis. eQTL data does, however, provide evidence implicating *GTDC1*. *GTDC1* encodes a glucosyltransferase whose expression is relatively high in peripheral blood leukocytes[36]. Chromosomal rearrangements of *MLL* (mixed lineage leukemia)

**Table 2 rs17481869 (2q22.3) genotypes and risk associated with BCP-ALL, high-hyperdiploid, and _ETV6-RUNX1_ childhood BCP-ALL subtypes**

| | RAF | | Number | | | | |
|---|---|---|---|---|---|---|---|
| **All BCP-ALL** | **Cases** | **Controls** | **Cases** | **Controls** | **OR** | **CI** | **P-value** |
| UK GWAS I | 0.08 | 0.07 | 824 | 5200 | 1.18 | (0.95–1.46) | $1.37 \times 10^{-1}$ |
| German GWAS | 0.10 | 0.08 | 834 | 2024 | 1.25 | (1.01–1.56) | $4.33 \times 10^{-2}$ |
| UK GWAS II | 0.10 | 0.07 | 784 | 7385 | 1.52 | (1.25–1.84) | $2.53 \times 10^{-5}$ |
| Combined | | | 2442 | 14,609 | 1.32 | (1.17–1.49) | $5.36 \times 10^{-6}$ |
| | | | | | | $P_{het} = 0.19$ | $I^2 = 39.3\%$ |
| **High-hyperdiploid** | | | | | | | |
| UK GWAS I | 0.06 | 0.07 | 289 | 5200 | 0.86 | (0.61–1.22) | $4.03 \times 10^{-1}$ |
| German GWAS | 0.08 | 0.08 | 176 | 2024 | 0.98 | (0.64–1.48) | $9.11 \times 10^{-1}$ |
| UK GWAS II | 0.10 | 0.07 | 251 | 7385 | 1.48 | (1.06–2.08) | $2.13 \times 10^{-2}$ |
| Combined | | | 716 | 14,609 | 1.10 | (0.89–1.35) | 0.38 |
| | | | | | | $P_{het} = 0.07$ | $I^2 = 62\%$ |
| **_ETV6-RUNX1_-positive** | | | | | | | |
| UK GWAS I | 0.11 | 0.07 | 126 | 5200 | 2.01 | (1.20–3.39) | $8.52 \times 10^{-3}$ |
| German GWAS | 0.12 | 0.08 | 63 | 2024 | 1.72 | (0.88–3.38) | $1.14 \times 10^{-1}$ |
| UK GWAS II | 0.13 | 0.07 | 220 | 7385 | 2.34 | (1.64–3.35) | $2.90 \times 10^{-6}$ |
| Combined | | | 409 | 14,609 | 2.14 | (1.64–2.80) | $3.20 \times 10^{-8}$ |
| | | | | | | $P_{het} = 0.70$ | $I^2 = 0\%$ |

Note: P-values for each individual study were generated using SNPTEST v2.5.2 software. Combined P-values and estimates were obtained using a fixed-effects model using beta values and standard errors. RAF risk allele frequency, OR odds ratio, $P_{het}$ P heterogeneity, $I^2$ index to quantify dispersion of odds ratio, CI confidence interval

genes are associated with infant leukemia and intriguingly _GTDC1_ has been identified as a 3′ _MLL_ fusion partner in acute leukemia[37].

Most cancer GWAS risk loci map to non-coding regions of the genome and in-so-far as they have been deciphered their functional basis has been attributed to changes in regulatory regions influencing gene expression[33,38,39]. The finding that the current and previously identified risk SNPs show a propensity to map within regions of B-cell active chromatin is consistent with such a model of disease susceptibility in ALL. It is therefore noteworthy that SMR analysis revealed significant relationships between 10p12.2 risk variants and _PIP4K2A_ expression and 10q26.13 risk variants and _FAM53B_ expression suggesting a mechanism for these associations.

Our analysis sheds further light on inherited predisposition to childhood ALL. Functional characterization of risk loci identified should provide additional insight into the biological and etiological basis of this malignancy. While the power of our meta-analysis to identify common variants loci (MAF > 0.2) associated with relative risks ≥ 1.2 was around 80%, we acknowledge that we had low power to detect alleles conferring more moderate effects or were present at low frequency. By inference, these types of variant may be responsible for a larger proportion of the heritable risk of ALL. Hence, a large number of risk SNPs may as yet be unidentified. Finally, as we have demonstrated, considering ALL subtypes individually should reveal additional specific risk variants.

## Methods

**Ethics.** The ascertainment patient samples and associated clinical information was conducted with informed consent according to ethical board approval. Specifically, ethical committee approval was obtained for Medical Research Council UKALL97/99 trial by UK therapy centers and approval for UKALL2003 from the Scottish Multi-Centre Research Ethics Committee (REC:02/10/052)[40,41]. Additionally ethical approval was granted by the Childhood Leukemia Cell Bank, the United Kingdom Childhood Cancer Study, and University of Heidelberg.

**Published GWAS samples.** The United Kingdom (UK) GWAS I and German GWAS have been previously published[6,7]. In summary, UK GWAS I comprised (numbers post quality control (QC)) 824 BCP-ALL cases (360 female, average age at diagnosis 5.5 years) genotyped using Human 317K arrays (Illumina, San Diego; http://www.illumina.com); control genotypes were obtained from 2699 individuals

from the 1958 British Birth Cohort (Hap1.2M-Duo Custom array data) and 2501 from the UK Blood Service produced by the Wellcome Trust Case Control Consortium 2 (http://www.wtccc.org.uk/; 51% male)[42]. The German GWAS comprised 1155 cases (620 male; mean age at diagnosis 6 years) from the Berlin–Frankfurt–Münster (BFM) trials (1993–2004) genotyped using Illumina Human OmniExpress-12v1.0 arrays (834 samples post QC). Control data was generated on 2132 (50% male) healthy individuals from the Heinz Nixdorf Recall study; 704 individuals genotyped using Illumina-HumanOmni1-Quad_v1 and 1428 individuals genotyped on Illumina-HumanOmniExpress-12v1.0 platform. In total 2024 controls remained post QC in the German cohort.

**New GWAS samples.** UK GWAS II consisted of 1021 BCP-ALL cases recruited to Medical Research Council UK ALL-2003 (2003–2011) (683 cases; 307 females, mean age: 5.9 years) and ALL-97/99 trials[40,41] (338 cases, 160 female, mean age: 4.9 years) obtained from the Bloodwise Childhood Leukemia Cell Bank (www.cellbank.org). DNA was extracted from cell pellets by standard ethanol precipitation methods. Samples were then genotyped on an Infinium OncoArray-500K BeadChip from Illumina comprising a 250K SNP genome-wide backbone and a 250K custom content selected across multiple consortia within COGS (Collaborative Oncological Gene-Environmental Study). OncoArray genotyping was carried out in accordance with the manufacturer's recommendations by the High-Throughput Genomics Group, Oxford Genomics Center. Prior to genotyping DNA samples were quantified by Quant-iT PicoGreen (Thermo Fisher Scientific, MA, USA), normalized and 50 ng/μl aliquots plated in 96 deep-well plates. Post QC we obtained genotype data for 784 cases (365 female; mean age at diagnosis 5.3 years). Controls consisted of: (1) 2976 cancer-free, men ascertained by the PRACTICAL Consortium; (2) 4446 cancer-free women from the UK through the Breast Cancer Association Consortium. All controls were genotyped on Infinium OncoArray-500K BeadChip arrays.

**Statistic and bioinformatics analysis of GWAS data sets.** Analyses and/or data management were undertaken using R v3.2.3 (R Core Team 2013; http://www.R-project.org/)[72], PLINK v1.9[43], and SNPTEST v2.5.2 software[44]. GenomeStudio software (Illumina, San Diego; Available at: http://www.illumina.com) was used to extract genotypes from raw data. QC of all GWAS data sets was performed as suggested by Anderson et al[45]. PLINK v1.9[43] was used for conducting the sample and SNP QC steps. Specifically, individuals with low call rate (<95%) as well as all individuals with non-European ancestry (using the HapMap version 2 CEU, JPT/CHB, and YRI populations as a reference) were excluded using the _smartpca_ package, part of EIGENSOFT v4.2[46,47]. SNPs with a call rate <95% were excluded as were those with a MAF < 0.01 or displaying significant deviation from Hardy–Weinberg equilibrium (i.e., $P < 10^{-5}$). The adequacy of case-control matching and possibility of differential genotyping of cases and controls were formally evaluated using QQ plots of test statistics. The inflation factor λ was calculated by dividing the median of the test statistics by the median expected values from a $\chi^2$ distribution with 1 degree of freedom. Q–Q plots were generated and inflation factors estimated using R. Uncorrected and pre imputation QQ plots of UK GWAS I, UK GWAS II, and German GWAS showed λ values of 1.01, 1.05, and 1.10, respectively. Prior to imputation the data sets were pre-phased by

estimating haplotypes from the GWAS data sets using Segmented HAPlotype Estimation and Imputation Tool to make imputation less computationally intensive[48,49]. Prediction of the untyped SNPs was carried out using IMPUTE v2.3.0 based on the data from the 1000 Genomes Project (Phase 1 integrated variant set, v3.20101123, http://www.1000genomes.org, 9 December 2013) and UK10K (ALSPAC, EGAS00001000090/EGAD00001000195, and TwinsUK, EGAS00001000108/EGAD00001000194, studies only; http://www.uk10k.org/) as reference. In order to account for genomic inflation post imputation in the German data set, eigenvectors were inferred using the "smartpca" component within EIGENSOFT v2.4 and adjustment was carried out by including the first two eigenvectors as covariates in SNPTEST during association analysis[46,47]. The inflation factor $\lambda$ and $\lambda_{1000}$ was again calculated for all SNPs post imputation, QC[13,50]. The association between each SNP and risk was calculated using SNPTEST assuming an additive model using a "-frequentist" test and applying a default genotype calling probability threshold of 0.9. Where applicable the first two eigenvectors were used as covariates in the association analyses for that data set. ORs and 95% CIs were obtained from the beta values and standard errors obtained from the SNPTEST output. Meta-analyses were performed using META v1.7[51] pooling the beta values and standard error for SNPs from each GWAS data sets. Association meta-analyses only included markers with info scores >0.8, imputed call rates/SNP >0.9, and MAFs > 0.01. Collectively the three GWAS provided genotype data on 2442 cases (mean age at diagnosis 5.6 years; 54% male) and 14,609 controls (45% male) with data for 6,755,715 SNPs[6,7,9]. We calculated Cochran's Q statistic to test for heterogeneity and the $I^2$ statistic to quantify the proportion of the total variation that was caused by heterogeneity[52].

LD metrics were calculated in PLINK[43] and vcftools[53] using UK10K genomic data. LD blocks were defined on the basis of HapMap recombination rate, as defined by using the Oxford recombination hotspots, and on the basis of distribution of CIs[54,55]. Association plots were generated using visPIG[14].

**HLA imputation**. Classical HLA alleles were imputed, both common and rare (A, B, C, DQA1, DQB1, DRB1) and coding variants across the HLA region using SNP2HLA[29]. The imputation was based on a reference panel from the T1DGC consisting of genotype data from 5225 individuals of European descent with genotyping data of 8961 common SNPs and indel polymorphisms across the HLA region, and four digit genotyping data of the HLA class I and II molecules. This reference panel has been used previously and showed high imputation quality for the HLA regions in other studies[27–29]. Individual GWAS studies were imputed at the 6p21 region and meta-analyzed to identify significant HLA risk alleles. A significance threshold of $5.7 \times 10^{-6}$ was set after Bonferroni correction as the number of SNPs tested was 8654.

**Sanger sequencing**. To assess the accuracy of imputed genotypes, a random series of samples was Sanger sequenced using BigDye® Terminator v3.1 Cycle Sequencing Kit (Life Technologies, CA, USA) and analyzed using a ABI 3700xl sequencer (Applied Biosystems, CA, USA). Oligonucleotide primer sequences are provided in Supplementary Table 12.

**Chromatin mark enrichment analysis**. To assess for an over-representation of markers for open chromatin the variant set enrichment method of Cowper-Sal Lari et al. was adapted[56]. For each risk locus, SNPs in LD were defined (i.e., $R^2 > 0.8$ and $D' > 0.8$), and termed associated variant set (AVS). Transcription factor ChIP-Seq broad peak data were obtained from the ENCODE project for 14 cell lines for H3K27ac, H3k4me1, and H3K4me3 chromatin signatures. ChIP-Seq broad peak data for three AML and six childhood ALL cell types were obtained from the Blue-Print Epigenome database (www.blueprint-epigenome.eu)[15]. For each mark, overlap of SNPs in the AVS and the ChIP peak were derived, generating a mapping score. The null hypothesis was tested by scoring randomly chosen SNPs with the same LD structure at the risk-associated SNPs. After 10,000 iterations, approximate P-values were calculated as the proportion of permutations where null mapping score was at least equal to the AVS mapping score. Enrichment was calculated normalizing scores to the median of the null model.

**Hi-C analysis**. Hi-C analysis was conducted using the HUGIn browser[57], which is based on the analysis by Schmitt et al[58]. Specifically we analyzed Hi-C data generated on the H1 ES Cells and GM12878 lymphoblastoid cell lines originally described in Dixon et al.[59] and Schmitt et al.[58], respectively. Plotted topologically associating domains boundaries were obtained from the insulating score method at 40 kb bin resolution[57]. We searched for significant interactions (P-values generated using "Fit-Hi-C"[18]) between bins overlapping the currently identified ALL risk loci with target genes (e.g., "virtual 4C").

**Functional annotation**. SNPs in LD ($r^2 > 0.8$) with the top SNPs from each risk loci were assessed for histone marks in relevant tissue, proteins bound and location were annotated using HaploReg[17] (Supplementary Data 1). eQTL analysis was performed by testing each sentinel SNP with genes 1MB upstream and downstream using the whole blood tissue data available from GTEx portal v6p[60] and Blood eQTL browser[61] (Supplementary Data 1). Methylation quantitative trait loci (mQTL) for all known BCP-ALL risk loci where assessed using the mQTL

Database (www.mqtldb.org), which shows the presence of significant methylated CpG sites at various stages of life as described by Gaunt et al[62].

**SMR analysis**. SMR analysis was conducted as per Zhu et al. (at http://cnsgenomics.com/software/smr/index.html)[63]. Publicly available eQTL data was extracted from the whole blood eQTL, Muther consortia, and GTEx16 v6p release portals[60,61,64]. GWAS summary statistics files were generated from the meta-analysis of UK GWAS I, UK GWAS II, and German GWAS data sets. Reference files were generated by merging 1000 genomes phase 3 and UK10K (ALSPAC and TwinsUK) vcfs. Summary eQTL files for the GTEx samples were generated from downloaded v6p "all_SNPgene_pairs" files. BESD files were generated from downloaded SNP-gene eQTL data, which were converted into a query flat file format as mentioned in the SMR online guide (http://cnsgenomics.com/software/smr) and then using the –make-besd command to make binary versions of the files. Only probes with eQTL $P < 5.0 \times 10^{-8}$ were considered in the SMR analysis. A threshold for the SMR test of $P_{smr} < 1.3 \times 10^{-4}$ corresponding to a Bonferroni correction for 38 tests for all the 23 genes within 1 MB of the sentinel risk SNPs in each risk loci (38 gene probes with a top eQTL $P < 5 \times 10^{-8}$). HEIDI test P-values < 0.05 were taken to indicate significant heterogeneity as suggested by Zhu et al. For the two genes passing the thresholds, plots of eQTL and GWAS associations as well as plots of GWAS and eQTL effect sizes were constructed.

**Relationship between SNP genotype and survivorship**. The relationship between SNP genotype and survival was analyzed in the, German AIEOP-BFM series, MRC ALL 97/99 and the UKALL2003 series. The German series consisted of 834 patients within the AIEOP-BFM 2000 trial[65]. Patients were treated with conventional chemotherapy (i.e., prednisone, vincristine, daunorubicin, l-aspar-aginase, cyclophosphamide, ifosfamide, cytarabine, 6-mercaptopurine, 6-thioguanine, and methotrexate), a subset of those with high-risk ALL were treated with cranial irradiation and/or stem cell transplantation. Events, for EFS, were defined as resistance to therapy, relapse, secondary cancer, or death. Kaplan–Meier methodology was used to estimate survival rates, with differences between groups tested using the log-rank method (two-sided P-values). Cumulative incidences of competing events were calculated using the methodology of Kalbfleisch and Prentice[66], and compared using Gray's test[67]. Cox regression analysis was used to estimate hazard ratios and 95% CIs adjusting for clinically relevant covariates.

The full details regarding the recruitment, classification, and treatment of patients on MRC ALL97/99 (1997–2002) or UKALL2003 (2003–2011) have been published[41,68–70]. In ALL97, patients were classified as standard or high risk based on the Oxford score. In ALL99 and UKALL2003, patients were initially assigned to regimen A or B based on whether they were NCI standard or high risk. Regimen A comprised a three drug induction followed by consolidation, CNS-directed therapy, interim maintenance, delayed intensification, and continuing therapy. Regimen B patients additionally received a four drug induction and BFM consolidation. Treatment response and cytogenetics were used to re-assign high-risk patients to regimen C to receive augmented BFM consolidation and Capizzi maintenance. In ALL99 and ALL2003, early treatment response was measured by marrow morphology at day 8/15 for regimen B/A-treated patients. In addition, ALL2003 patients were randomized to regimen C if their end of induction minimal residual disease levels—evaluated by real-time quantitative PCR analysis of immunoglobulin and T-cell receptor gene rearrangements—were >0.01%. Survival analysis considered two endpoints: EFS defined as time to relapse, second tumor or death, censoring at last contact; and relapse rate defined as time to relapse for those achieving a complete remission, censoring at death in remission or last contact. Survival rates were calculated and compared using Kaplan–Meier methods and log-rank tests. All analyses were performed using Intercooled Stata 13.0 (Stata Corporation, USA).

**Contribution of genetic variance to familial risk**. Estimation of risk variance associated with each SNP was performed as per Pharoah et al[71]. For an allele (i) of frequency p, relative risk R and log risk r, the risk distribution variance ($V_i$) is:
$$V_i = (1-p)^2 E^2 + 2p(1-p)(r-E)^2 + p^2(2r-E)^2,$$
where E is the expected value of r given by:
$$E = 2p(1-p)r + 2p^2 r$$
For multiple risk alleles the distribution of risk in the population tends toward the normal with variance:
$$V = \Sigma V_i$$
The percentage of total variance was calculated assuming a familial risk of childhood ALL of 3.2 (95% CI 1.5–5.9) as per Kharazmi et al[4]. All genetic variance (V) associated with susceptibility alleles is given as $\sqrt{3.2}$[4]. The proportion of genetic risk attributable to a single allele is:
$$V_i / V$$
Eleven risk loci were included in the calculation of the PRS for childhood ALL by selecting the top SNP from the current meta-analysis from each previously published loci in addition to the two risk loci discovered in this study. The eleven variants are thought to act independently as previous studies have shown no interaction between risk loci[6–8]. PRS were generated as per Pharoah et al. assuming a log-normal distribution $LN(\mu, \sigma^2)$ with mean $\mu$, and variance $\sigma^2$[32]. The population $\mu$ was set to $\sigma^2/2$, in order that the overall mean PRS was 1.0. The

sibling relative risk were assumed to be 3.2[4]. The discriminatory value of risk SNPs was examined by determining the AUC for the ROC curve.

**GCTA to estimate heritability**. Since artefactual differences in allele frequencies between cases and controls have the potential to bias estimation genetic variation, additional QC measures were imposed on the GWAS data sets which have been advocated by Lee et al[73]. Typed SNPs were excluded if they had a MAF < 0.01 or a HWE test with $P < 0.05$. SNPs were also excluded if a differential missingness test between cases and controls was $P < 0.05$. In addition, individuals were excluded if having a relatedness score of >0.05. Filtering resulted in the 260,127 SNPs in the UK GWAS I and 355,899 SNPs in UK GWAS II data sets, respectively. GCTA (http://cnsgenomics.com/software/gcta/) was employed to estimate the fraction of the phenotypic variance attributed by SNPs given a prevalence of 0.0005 for ALL[30].

**Data availability**. The UK GWAS I control set comprised 2699 individuals in the 1958 British Birth Cohort (Hap1.2M-Duo Custom array data) and 2501 individuals from the UK Blood Service obtained from the publicly accessible data generated by the Wellcome Trust Case Control Consortium 2 (http://www.wtccc.org.uk/; WTCCC2:EGAD00000000022, EGAD00000000024). The reference panels used in the imputation can be obtained from the 1000 genomes phased haplotypes ($n = $ 1092) from the Phase I integrated variant set release (ftp://ftp.1000genomes.ebi.ac.uk/vol1/ftp/release/20110521/) and the UK10K ($n = 3781$; EGAS00001000090, EGAD00001000195, EGAS00001000108; www.uk10k.org) sequenced data sets. eQTL data for various functional analyses were obtained from the MuTHer studies (genome-wide expression profiled samples with genotype array data and methylation data; E-TABM-1140), Blood eQTL (whole-genome gene exression array data sets with RNA sequencing and genotyping data: E-TABM-1036, E-MTAB-945, E-MTAB-1708; http://www.nature.com/ng/journal/v45/n10/abs/ng.2756.html), and ENCODE transcription factor binding data sets (transcription factor ChIP-seq data from various tissues: http://genome.ucsc.edu/ENCODE/downloads.html). ChIP-seq broad peak data for childhood ALL and AML cells were obtained from the BluePrint Epigenome (dcc.blueprint-epigenome.eu) for samples S00FGCH1, S005GFH1, S00KPBH1, S017E3H1, S0179DH1, S01GRFH1, S01GQHH1, S0176JH1, and S0177HH1. The UK GWAS II data set can be accessed through the European Genome-Phenome Archive website (EGA, https://ega-archive.org) under the study accession EGAS00001002809. All other relevant data are available on request to the authors.

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

## Author contributions

R.S.H. obtained financial support for the new GWAS. R.S.H. designed the study and drafted the manuscript along with J.V. and J.S. J.V. performed the bioinformatics and statistical analysis, GWAS sample preparation, and validation genotyping. J.V. and P.B. supervised and coordinated the genotyping of the new GWAS samples. A.H. supported in sample DNA extraction. J.V. and B.K. performed the SMR eQTL analyses. J.V. and P.J. L. performed the transcription factor enrichment analysis. J.M.A., C.J.H., A.V.M., E.R., S.E.K., E.S., P.D.T., M.G., and J.A.I. contributed toward the UKCCS samples used in the UK GWAS I. R.K., P.H., M.M.N., S.H.-H., K.-H.J. contributed toward the Heinz-Nixdorf control data set. C.R.B., M.St., M.Sc., K.H., R.K., and S.R. provided samples for the German GWAS. The PRACTICAL consortium, D.E., P.P., A.D., J.P., F.C., A.S., R.E., Z.K.-J., K.M., and N.P. provided control samples for the UK GWAS II. M.St. and M.Z. conducted the survival analysis in the German GWAS. A.V.M. and A.V. conducted the survival analysis in the UK GWAS II series. All authors contributed toward the final paper.

## Additional information

**Competing interests:** The authors declare no competing financial interests.

## The PRACTICAL Consortium

Brian E. Henderson[28], Christopher A. Haiman[28], Sara Benlloch[29,30], Fredrick R. Schumacher[31,32], Ali Amin Al Olama[29,33], Sonja I. Berndt[34], David V. Conti[28], Fredrik Wiklund[35], Stephen Chanock[34], Victoria L. Stevens[36], Catherine M. Tangen[37], Jyotsna Batra[38,39], Judith Clements[38,39], Henrik Gronberg[35], Johanna Schleutker[40,41,42], Demetrius Albanes[34], Stephanie Weinstein[34], Alicja Wolk[43], Catharine West[44], Lorelei Mucci[45], Géraldine Cancel-Tassin[46,47], Stella Koutros[34], Karina Dalsgaard Sorensen[48,49],

Lovise Maehle[50], David E. Neal[51,52], Ruth C. Travis[53], Robert J. Hamilton[54], Sue Ann Ingles[28], Barry Rosenstein[55,56], Yong-Jie Lu[57], Graham G. Giles[58,59], Adam S. Kibel[60], Ana Vega[61], Manolis Kogevinas[62,63,64,65], Kathryn L. Penney[66], Jong Y. Park[67], Janet L. Stanford[68,69], Cezary Cybulski[70], Børge G. Nordestgaard[71,72], Hermann Brenner[73,74,75], Christiane Maier[76], Jeri Kim[77], Esther M. John[78,79], Manuel R. Teixeira[80,81], Susan L. Neuhausen[82], Kim De Ruyck[83], Azad Razack[84], Lisa F. Newcomb[68,85], Davor Lessel[86], Radka Kaneva[87], Nawaid Usmani[88,89], Frank Claessens[90], Paul A. Townsend[91], Manuela Gago-Dominguez[92,93], Monique J. Roobol[94] & Florence Menegaux[95]

[28]Department of Preventive Medicine, Keck School of Medicine, University of Southern California/Norris Comprehensive Cancer Center, Los Angeles, CA 90033, USA. [29]Department of Public Health and Primary Care, Centre for Cancer Genetic Epidemiology, Strangeways Research Laboratory, University of Cambridge, Cambridge CB2 0SP, UK. [30]The Institute of Cancer Research, London SM2 5NG, UK. [31]Department of Epidemiology and Biostatistics, Case Western Reserve University, Cleveland, OH 44106, USA. [32]Seidman Cancer Center, University Hospitals, Cleveland, OH 44106, USA. [33]Department of Clinical Neurosciences, University of Cambridge, Cambridge CB2 2PY, UK. [34]Division of Cancer Epidemiology and Genetics, National Cancer Institute, NIH, Bethesda, MD 20814, USA. [35]Department of Medical Epidemiology and Biostatistics, Karolinska Institute, Stockholm 171 77, Sweden. [36]Epidemiology Research Program, American Cancer Society, 250 Williams Street, Atlanta, GA 30303, USA. [37]SWOG Statistical Center, Fred Hutchinson Cancer Research Center, Seattle, WA 98109-1024, USA. [38]Australian Prostate Cancer BioResource (APCB), Australian Prostate Cancer Research Centre-Qld, Institute of Health and Biomedical Innovation and School of Biomedical Science, Queensland University of Technology, Brisbane 4001 QLD, Australia. [39]Translational Research Institute, Brisbane 4102 QLD, Australia. [40]Department of Medical Biochemistry and Genetics, Institute of Biomedicine, University of Turku, Turku FI-20014, Finland. [41]Tyks Microbiology and Genetics, Department of Medical Genetics, Turku University Hospital, Turku 20521, Finland. [42]BioMediTech, University of Tampere, Tampere 33100, Finland. [43]Division of Nutritional Epidemiology, Institute of Environmental Medicine, Karolinska Institutet, Solna SE-171 77, Sweden. [44]Institute of Cancer Sciences, University of Manchester, Manchester Academic Health Science Centre, Radiotherapy Related Research, The Christie Hospital NHS Foundation Trust, Manchester M13 9PL, UK. [45]Department of Epidemiology, Harvard School of Public Health, Boston, MA 02115, USA. [46]CeRePP, Pitie-Salpetriere Hospital, Paris 75020, France. [47]UPMC Univ Paris 06, GRC N°5 ONCOTYPE-URO, CeRePP, Tenon Hospital, Paris 75020, France. [48]Department of Molecular Medicine, Aarhus University Hospital, Aarhus DK-8200, Denmark. [49]Department of Clinical Medicine, Aarhus University, Aarhus 8000, Denmark. [50]Department of Medical Genetics, Oslo University Hospital, Oslo 0424, Norway. [51]Department of Oncology, Addenbrooke's Hospital, University of Cambridge, Cambridge CB2 0QQ, UK. [52]Li Ka Shing Centre, Cancer Research UK Cambridge Research Institute, Cambridge CB2 0RE, UK. [53]Cancer Epidemiology, Nuffield Department of Population Health, University of Oxford, Oxford OX3 7LF, UK. [54]Department of Surgical Oncology, Princess Margaret Cancer Centre, Toronto M5G 2C4, Canada. [55]Department of Radiation Oncology, Icahn School of Medicine at Mount Sinai, New York, NY 10029-5674, USA. [56]Department of Genetics and Genomic Sciences, Icahn School of Medicine at Mount Sinai, New York, NY 10029-5674, USA. [57]Centre for Molecular Oncology, Barts Cancer Institute, John Vane Science Centre, Queen Mary University of London, London EC1M 6BQ, UK. [58]Cancer Epidemiology Centre, The Cancer Council Victoria, Melbourne 3004 VIC, Australia. [59]Centre for Epidemiology and Biostatistics, Melbourne School of Population and Global Health, The University of Melbourne, Melbourne 3010, Australia. [60]Division of Urologic Surgery, Brigham and Womens Hospital, Boston, MA 02115, USA. [61]Fundación Pública Galega de Medicina Xenómica-SERGAS, Grupo de Medicina Xenómica, CIBERER, IDIS, Santiago de Compostela 15706, Spain. [62]Centre for Research in Environmental Epidemiology (CREAL), Barcelona Institute for Global Health (ISGlobal), Barcelona 08036, Spain. [63]CIBER Epidemiología y Salud Pública (CIBERESP), Madrid 28029, Spain. [64]IMIM (Hospital del Mar Research Institute), Barcelona 08003, Spain. [65]Universitat Pompeu Fabra (UPF), Barcelona 08005, Spain. [66]Channing Division of Network Medicine, Department of Medicine, Brigham and Women's Hospital/Harvard Medical School, Boston, MA 02115, USA. [67]Department of Cancer Epidemiology, Moffitt Cancer Center, Tampa 33612, USA. [68]Division of Public Health Sciences, Fred Hutchinson Cancer Research Center, Seattle, WA 98109, USA. [69]Department of Epidemiology, School of Public Health, University of Washington, Seattle, WA 98195, USA. [70]International Hereditary Cancer Center, Department of Genetics and Pathology, Pomeranian Medical University, Szczecin 70-204, Poland. [71]Faculty of Health and Medical Sciences, University of Copenhagen, Copenhagen DK-2200, Denmark. [72]Department of Clinical Biochemistry, Herlev and Gentofte Hospital, Copenhagen University Hospital, Herlev 2730, Denmark. [73]Division of Clinical Epidemiology and Aging Research, German Cancer Research Center (DKFZ), Heidelberg 69120, Germany. [74]German Cancer Consortium (DKTK), German Cancer Research Center (DKFZ), Heidelberg 69120, Germany. [75]Division of Preventive Oncology, German Cancer Research Center (DKFZ) and National Center for Tumor Diseases (NCT), Heidelberg 69120, Germany. [76]Institute for Human Genetics, University Hospital Ulm, Ulm 89081, Germany. [77]Department of Genitourinary Medical Oncology, The University of Texas MD Anderson Cancer Center, Houston, TX 77030, USA. [78]Cancer Prevention Institute of California, Fremont, CA 94538, USA. [79]Department of Health Research & Policy (Epidemiology) and Stanford Cancer Institute, Stanford University School of Medicine, Stanford, CA 94305-5456, USA. [80]Department of Genetics, Portuguese Oncology Institute of Porto, Porto 4200-072, Portugal. [81]Biomedical Sciences Institute (ICBAS), University of Porto, Porto 4050-313, Portugal. [82]Department of Population Sciences, Beckman Research Institute of the City of Hope, Duarte, CA 91010, USA. [83]Faculty of Medicine and Health Sciences, Basic Medical Sciences, Ghent University, Gent 9000, Belgium. [84]Department of Surgery, Faculty of Medicine, University of Malaya, Kuala Lumpur 50603, Malaysia. [85]Department of Urology, University of Washington, Seattle, WA 98195, USA. [86]Institute of Human Genetics, University Medical Center Hamburg-Eppendorf, Hamburg 20246, Germany. [87]Molecular Medicine Center, Department of Medical Chemistry and Biochemistry, Medical University, Sofia 1431, Bulgaria. [88]Department of Oncology, Cross Cancer Institute, University of Alberta, Edmonton T6G 1Z2 AB, Canada. [89]Division of Radiation Oncology, Cross Cancer Institute, Edmonton T6G 2R7 AB, Canada. [90]Department of Cellular and Molecular Medicine, Molecular Endocrinology Laboratory, KU Leuven, Leuven 3000, Belgium. [91]Institute of Cancer Sciences, Manchester Cancer Research Centre, University of Manchester, Manchester Academic Health Science Centre, St. Mary's Hospital, Manchester M13 9PL, UK. [92]Genomic Medicine Group, Galician Foundation of Genomic Medicine, Instituto de Investigacion Sanitaria de Santiago de Compostela (IDIS), Complejo Hospitalario Universitario de Santiago, Servicio Galego de Saúde, SERGAS, Santiago de Compostela 15706, Spain. [93]Moores Cancer Center, University of California San Diego, La Jolla, CA 92093, USA. [94]Department of Urology, Erasmus University Medical Center, Rotterdam 3015, The Netherlands. [95]Cancer & Environment Group, Center for Research in Epidemiology and Population Health (CESP), INSERM, University Paris-Sud, University Paris-Saclay, Villejuif 94800, France

