## [Peer Review File · Nature Communications]

Reviewers' comments:

Reviewer #1 (Remarks to the Author):

The authors performed a meta-analysis of two existing GWAS and a new GWAS of B-cell precursor acute lymphoblastic leukemia (BCP-ALL) with a total of 2,442 cases and 14,609 controls. New risk loci for BCP-ALL at 8q24.21 and ETV6-RUNX1 fusion-positive lymphoblastic leukemia at 2q22.3 were reported in this study.

The results seems interesting, however, I have some major comments, and a few minor comments.

Major comments

1. Quantile-quantile plots of SNPs showed that a lot of variants were with extreme low P-values in each study ($P < 1 \times 10^{-20}$), however, the M-plot of final meta-results is relative clear and the most significant variants were only near $P = 1 \times 10^{-20}$ (Figure 1); so I think the authors should do some quality controls after meta-analysis, please describe the criteria clearly in the article.
2. The quantile-quantile plot for UK GWAS II showed an obvious deviation from expected P-values; please provide the exact λ , or $\lambda 1000$ other than $\lambda 100$, for each study and the final meta-results.
3. According to Figure 1 and Supplementary Table 10, rs7449087 at chr.5 was also significantly associated with BCP-ALL risk at $P < 5 \times 10^{-8}$, why it is not mentioned in the main text?
4. Has HLA allele been reported to be associated with the risk of BCP-ALL? Why the HLA-allele is discussed separately here?
5. Different region intervals were used in this study, such as 1Mb in the SMR analysis (page 6) and 2Mb in Supplementary Table 10; as for SNP-expression analysis, which one should be used? Please provide basis and evidences.
6. As discussed in the manuscript, there probably be variability in the genetic effects on BCP-ALL risk by subtype; then I would like to know the heritable risk of different subtypes of BCP-ALL, please explain this.
7. MYC and GTDC1 were considered to be the susceptibility genes in the two identified loci through eQTL analysis, please provide more evidence about the regulatory mechanisms of the identified SNPs and susceptibility genes through functional experiments, such as Dual-Luciferase Reporter Assay System.
8. To test the potential clinical utility for risk profiling of BCP-ALL, a genetic risk was calculated; however, in practice I would like to know the areas under the ROC curve (AUC) in different study based on the eleven identified loci.

Minor comments

1. The abbreviations should be in a more standard way. In page 4, it is better to revise the B-cell precursor (BCP) ALL as B-cell precursor lymphoblastic leukemia (BCP-ALL); and then we should distinct BCP-ALL from ALL accordingly.
2. In figure 2, the reported SNP should be highlighted. It can be highlighted with a different color or with arrows.
3. It is better to optimize the structure of the manuscript to make it clear and readable. The grammar, style, structure and punctuation should be revised.

Reviewer #2 (Remarks to the Author):

Dr. Houlston and colleagues have identified a novel locus on chromosome 8q24 for B-cell precursor acute lymphoblastic leukemia (BCL-ALL, n=2442) and a novel locus on chromosome 2q22.3 for a subtype of ALL (ETV6-RUNX1 fusion-positive ALL, n=409) through a meta-analysis of three genome-wide association studies. The study design, experiments, and analyses are of high-quality. This is a well-written paper.

Although this is the largest study to date for BCL-ALL, the findings are a bit limited in significance with only the two loci identified. Moreover, the findings were not validated in an independent sample separate from the discovery GWAS samples used to identify the loci. Note, however, the authors did technically validate their findings using a different technology and the findings in Tables 1 and 2 are fairly consistent across the three GWASs. Finally, it would be nice to show the age and sex distribution of the cases and controls across the studies.

Reviewer #1 (Remarks to the Author):

The authors performed a meta-analysis of two existing GWAS and a new GWAS of B-cell precursor acute lymphoblastic leukemia (BCP-ALL) with a total of 2,442 cases and 14,609 controls. New risk loci for BCP-ALL at 8q24.21 and ETV6-RUNX1 fusion-positive lymphoblastic leukemia at 2q22.3 were reported in this study.

The results seems interesting, however, I have some major comments, and a few minor comments.

Major comments

1. Quantile-quantile plots of SNPs showed that a lot of variants were with extreme low P-values in each study ($P < 1 \times 10^{-20}$), however, the M-plot of final meta-results is relative clear and the most significant variants were only near $P = 1 \times 10^{-20}$ (Figure 1); so I think the authors should do some quality controls after meta-analysis, please describe the criteria clearly in the article.

Response: There are indeed very extreme P-values for associations at 7p12.2, 9p21.3 and 10q21.2. In the original Figure 1 we presented the y-axis terminated at $P = 1 \times 10^{-20}$, purely for presentational reasons. We appreciate that in retrospect this was an error and we have now revised Figure 1. Specifically, we have introduced a break in the Y-axis so as to display P-values $< 10^{-60}$.

2. The quantile-quantile plot for UK GWAS II showed an obvious deviation from expected P-values; please provide the exact λ , or λ_{1000} other than λ_{100} , for each study and the final meta-results.

Response: λ_{100} in the original manuscript signified the λ value corresponding to all SNPs. We apologise for generating this ambiguity. In the revised MS we now simply referred to this as λ . We additionally provide λ_{1000} values and the values for the final meta-analysis as the reviewer requested.

3. According to Figure 1 and Supplementary Table 10, rs7449087 at chr.5 was also significantly associated with BCP-ALL risk at $P < 5 \times 10^{-8}$, why it is not mentioned in the main text?

Response: We now make further reference to this SNP. Specifically, we state that the fidelity of imputation of SNP rs7449087 was poor ($r^2 = 0.81$) with no correlated directly typed SNP with P-value $< 1 \times 10^{-6}$, hence we did not consider this represented a bona fide association (Supplementary Table 4).

4. Has HLA allele been reported to be associated with the risk of BCP-ALL? Why the HLA-allele is discussed separately here?

Response: We now state "A relationship between variation within the major histocompatibility complex (MHC) region and risk of ALL has long been speculated. However, most studies have failed to address the complex LD patterns within the MHC or issues relating to population stratification. In view of the inconsistencies and limitations of published studies we conducted a more rigorous analysis. "

5. Different region intervals were used in this study, such as 1Mb in the SMR analysis (page 6) and 2Mb in Supplementary Table 10; as for SNP-expression analysis, which one should be used? Please provide basis and evidences.

Response: Our SMR analysis was based on a 2Mb window as generally advocated. We now state “Following from SMR analysis we also investigated whether the most strongly associated lead SNP at each risk locus, individually, was associated with the expression of genes within a 2MB window to ensure capture of long range interactions.”

6. As discussed in the manuscript, there probably be variability in the genetic effects on BCP-ALL risk by subtype; then I would like to know the heritable risk of different subtypes of BCP-ALL, please explain this.

Response: As requested these data are now provided.

7. MYC and GTDC1 were considered to be the susceptibility genes in the two identified loci through eQTL analysis, please provide more evidence about the regulatory mechanisms of the identified SNPs and susceptibility genes through functional experiments, such as Dual-Luciferase Reporter Assay System.

Response: To address this point we now state “Since chromatin looping interactions are fundamental for regulation of gene expression, we interrogated physical interactions at respective genomic regions defined by rs28665337 and rs17481869 in GM12878 lymphoblastoid and H1 human embryonic stem (ES) cells using Hi-C data. With the limitations of data from which may not fully reflect ALL biology, the regions containing rs28665337 and rs17481869 show significant chromatin looping interactions with the promotor regions of *MYC* and *GTDC1* respectively (Supplementary Fig.7)”

We would assert that additional work is outside the remit of the present analysis; a view which we believe is supported by the recent Nat Commun paper “Chang et al. Nat Commun. 2017 Sep 18;8(1):569. Common variants in MMP20 at 11q22.2 predispose to 11q deletion and neuroblastoma risk.”

8. To test the potential clinical utility for risk profiling of BCP-ALL, a genetic risk was calculated; however, in practice I would like to know the areas under the ROC curve (AUC) in different study based on the eleven identified loci.

Response: We now present a ROC curve (AUC) which yields a value of 0.73.

Minor comments

1. The abbreviations should be in a more standard way. In page 4, it is better to revise the B-cell precursor (BCP) ALL as B-cell precursor lymphoblastic leukemia (BCP-ALL); and then we should distinct BCP-ALL from ALL accordingly.

Response: Text revised as requested.

2. In figure 2, the reported SNP should be highlighted. It can be highlighted with a different color or with arrows.

Response: As requested Figure 2 has been revised.

3. It is better to optimize the structure of the manuscript to make it clear and readable. The grammar, style, structure and punctuation should be revised.

Response: We have hopefully revised the text to address such failings.

Reviewer #2 (Remarks to the Author):

Dr. Houlston and colleagues have identified a novel locus on chromosome 8q24 for B-cell precursor acute lymphoblastic leukemia (BCL-ALL, n=2442) and a novel locus on chromosome 2q22.3 for a subtype of ALL (ETV6-RUNX1 fusion-positive ALL, n=409) through a meta-analysis of three genome-wide association studies. The study design, experiments, and analyses are of high-quality. This is a well-written paper.

Response: We appreciate that the reviewer found our paper of interest.

Although this is the largest study to date for BCL-ALL, the findings are a bit limited in significance with only the two loci identified. Moreover, the findings were not validated in an independent sample separate from the discovery GWAS samples used to identify the loci. Note, however, the authors did technically validate their findings using a different technology and the findings in Tables 1 and 2 are fairly consistent across the three GWASs. Finally, it would be nice to show the age and sex distribution of the cases and controls across the studies.

Response: As requested we now provide the age and sex distribution of the cases and controls across the studies.

Reviewers' comments:

Reviewer #1 (Remarks to the Author):

The authors provided detailed responses that answered each query, as well as a much improved manuscript. We have no more questions.

Reviewer #2 (Remarks to the Author):

The authors didn't address my prior comment about using an independent sample set of cases and controls outside of the GWAS discovery set to validate their findings.